# Activation and evasion of type I interferon responses by SARS-CoV-2

Xiaobo Lei[1,2,3,5], Xiaojing Dong[1,5], Ruiyi Ma[1,5], Wenjing Wang[1], Xia Xiao[1], Zhongqin Tian[1], Conghui Wang[1], Ying Wang[1], Li Li[1], Lili Ren[1,2,3], Fei Guo[1], Zhendong Zhao[1], Zhuo Zhou[4✉], Zichun Xiang[1,2,3✉] & Jianwei Wang [1,2,3✉]

The pandemic of COVID-19 has posed an unprecedented threat to global public health. However, the interplay between the viral pathogen of COVID-19, SARS-CoV-2, and host innate immunity is poorly understood. Here we show that SARS-CoV-2 induces overt but delayed type-I interferon (IFN) responses. By screening 23 viral proteins, we find that SARS-CoV-2 NSP1, NSP3, NSP12, NSP13, NSP14, ORF3, ORF6 and M protein inhibit Sendai virus-induced IFN-β promoter activation, whereas NSP2 and S protein exert opposite effects. Further analyses suggest that ORF6 inhibits both type I IFN production and downstream signaling, and that the C-terminus region of ORF6 is critical for its antagonistic effect. Finally, we find that IFN-β treatment effectively blocks SARS-CoV-2 replication. In summary, our study shows that SARS-CoV-2 perturbs host innate immune response via both its structural and nonstructural proteins, and thus provides insights into the pathogenesis of SARS-CoV-2.

[1] NHC Key Laboratory of System Biology of Pathogens, Institute of Pathogen Biology, Chinese Academy of Medical Sciences and Peking Union Medical College, 100730 Beijing, P.R. China. [2] Key Laboratory of Respiratory Disease Pathogenomics, Chinese Academy of Medical Sciences and Peking Union Medical College, 100730 Beijing, P.R. China. [3] Christophe Merieux Laboratory, Institute of Pathogen Biology, Chinese Academy of Medical Sciences and Peking Union Medical College, 100730 Beijing, P.R. China. [4] Biomedical Pioneering Innovation Center, Beijing Advanced Innovation Center for Genomics, Peking University Genome Editing Research Center, School of Life Sciences, Peking University, 100871 Beijing, China. [5] These authors contributed equally: Xiaobo Lei, Xiaojing Dong, Ruiyi Ma. ✉email: zhouzhuo@pku.edu.cn; xiangzch@163.com; wangjw28@163.com

A novel human coronavirus (SARS-CoV-2) emerged in December 2019 and caused 6,931,000 cases, including 400,857 deaths involved 185 countries, areas, or territories as of June 8, 2020 (https://covid19.who.int/), posing a huge threat to global public health. The World Health Organization (WHO) has announced COVID-19 as a pandemic on 11 March 2020 (www.who.int/emergencies/diseases/novel-coronavirus-2019/events-as-they-happen). SARS-CoV-2 belongs to the *Coronaviridae* family, *Orthocoronavirinae* subfamily, *Betacoronaviruses* genus, *Sarbecovirus* subgenus[1]. After SARS-CoV, SARS-CoV-2 is the second virus that originated from bats and could infect human beings of *Sarbecovirus*[1–3]. The pathogenesis of SARS-CoV-2 remains largely unknown.

Similar to other viruses in *Sarbecovirus*, the genome of SARS-CoV-2 is approximately 29.7 kb long with a short untranslated region (UTR) in 5′ and 3′ terminus[1,3]. The SARS-CoV-2 genome encodes spike (S), envelope (E), membrane (M), nucleocapsid (N) proteins, accessory proteins 3, 6, 7a, 7b, 8, and 9b, and comprises a large open reading frame (ORF) encoding 1ab[1], which is further cleaved into 15 nonstructural proteins (NSP1–10, 12–16) by its papain-like proteinase (NSP3) and 3C-like proteinase (NSP5)[4]. Different from SARS-CoV, the SARS-CoV-2 genome encodes complete ORF8 but no 8b. The genome of SARS-CoV-2 has 79% nucleotide identity with that of SARS-CoV[1].

Innate immunity is the first line of host defense against virus infections and is initiated by recognition of pathogen-associated molecular patterns (PAMPs) via host pattern recognition receptors (PRRs)[5–8]. The type-I interferon system is a vital part of the innate immune response. The double-strand RNA (dsRNA), which is generated during coronavirus genome replication and transcription[9,10], could be recognized by the RIG-I-like receptors (RLRs), including the retinoic acid-inducible gene I (RIG-I) and/or melanoma differentiation gene 5 (MDA5) in the cytoplasm[11,12], or by toll-like receptors (TLRs) in the endosome[13,14]. The two caspase activation recruitment domains (CARD) of RIG-I and MDA5 could interact with the adapter mitochondrial antiviral signaling protein (MAVS, also termed as IPS-1, VISA, and Cardif)[15–18], which subsequently recruits the two IKK-related kinases, TANK-binding kinase 1 (TBK1) and inducible IκB kinase (IKKi), both of which phosphorylate interferon regulatory factor 3/7 (IRF3/7)[19]. After phosphorylation and dimerization, IRF3/7 translocates to the nucleus to activate the expression of IFN-α/β[20,21]. Concomitantly, MAVS recruits TANK1 by TRAF6 and activates the NF-κB signaling pathway, which could promote the cytokines production[15]. Alternatively, PAMPs could be recognized by Toll-like receptors (TLRs), and the downstream adapter proteins TRIF or MyD88 could signal to induce cytokines and chemokines production[13,22,23].

Once secreted, IFN-α/β function as autocrine and paracrine factors to induce the expression of IFN-stimulated genes (ISGs) via the Janus activated kinase (Jak)-signal transducers and activators of transcription (STAT1) signaling pathways. ISGs are the main effectors of IFN-mediated antiviral responses[24]. Accordingly, viruses have developed diverse strategies to counteract interferon responses. Coronavirus such as SARS-CoV could inhibit interferon production by (1) avoiding being recognized by PRRs[25,26], (2) compromising RIG-I or TLRs signaling[27], and (3) impeding IRF3 activation[28]. Moreover, SARS-CoV could antagonize the signaling pathway downstream of IFN production by blocking the nuclear translocation of STAT1[29]. However, whether and how SARS-CoV-2 antagonizes IFN production and signaling is not clear.

Several SARS-CoV proteins have been identified as IFN antagonists. The accessory open reading frames 6 (ORF6) play a critical role in counteracting host antiviral response and viral replication[30]. ORF6 suppresses the Sendai virus (SeV)-mediated IFN induction by inhibiting the phosphorylation and nuclear translocation of IRF3[31]. ORF6 also inhibits the STAT1 nuclear translocation without affecting its phosphorylation[31], possibly by interacting with KPNA2, which mediates the KPNAB binding to ORF6-KPNA2 complex[29].

Here, we investigate the interaction between SARS-CoV-2 and host antiviral responses. We demonstrate that host innate immune response is targeted by multiple viral proteins, among which ORF6 potently perturbs signaling pathways both upstream and downstream of IFN production. Moreover, we show that SARS-CoV-2 is susceptible to IFN treatment. Our results provide mechanistic insights into interactions between SARS-CoV-2 and the host.

## Results

**SARS-CoV-2 induces substantial but delayed IFN-β production.** To understand the interaction between SARS-CoV-2 and the host antiviral response, we firstly examined whether SARS-CoV-2 infection induces the expression of IFN-β and IFN-inducible genes 56 (ISG56). Calu-3, an airway epithelial cell line, were mock-infected or infected with SARS-CoV-2. In parallel, cells were infected or transfected with SeV and poly (I:C), respectively, both of which are often used to stimulate antiviral signaling pathways. At different time points postinfection (hpi), cells were harvested for determining host and viral RNA levels by quantitative PCR analysis, and the supernatants were collected and subjected to $TCID_{50}$ assays for measuring viral titers. SARS-CoV-2 replicated well in the Calu-3 cells, as seen from the increased number of viral transcripts and replicative viruses with prolonged infection time (Fig. 1a, b). Upon SARS-CoV-2 infection, the expressions of IFN-β and ISG56 are only marginally elevated until 12 hpi but are dramatically induced at 24 hpi (Fig. 1c, d). In contrast, SeV infection stimulates the expression of IFN-β and ISG56 as early as 4 hpi and peaked at 8 hpi, even though SeV RNA is much less produced compare to SARS-CoV-2 (Fig. 1e–g). Moreover, poly (I:C) induced IFN-β and ISG56 expression in a similar kinetic pattern to that detected in SeV infection (Fig. 1h, i). These observations show that SARS-CoV-2 infection stimulates substantial but delayed IFN production, suggesting that SARS-CoV-2 infection attenuated host antiviral response.

**SARS-CoV-2 proteins interfere with IFN-β activation.** To explore which proteins of SARS-CoV-2 could regulate the innate immune responses, we cloned SARS-CoV-2 genes after codon optimization, including nonstructural genes *NSP1-10, NSP12–16*, structural genes *S, E, M*, and *N*, and accessory protein genes *ORF3, ORF6, ORF7a*, and *ORF8* (Fig. 2a). ORF7b was not included due to the small size of 43 amino acids (a.a.). An NSP3 fragment (nucleotide sequence 2250–3183), which encode the papain-like protease 2 domain, was cloned due to the difficulties in synthesizing the full-length NSP3 of 5835 bp. Western blot showed that all genes could be expressed, albeit at different levels (Fig. 2b). Next, we evaluated the effect of individual SARS-CoV-2 proteins on IFN-β promoter activation. 293T cells were transiently transfected with the vector plasmid or with plasmids expressing SARS-CoV-2 proteins, along with an IFN-β promoter-driven luciferase reporter plasmid (pIFN-β-Luc) and a control pRL-TK plasmid. After 24 h, cells were stimulated with SeV for 12 h, and the luciferase activity was determined. We found that SARS-CoV-2 proteins exerted divergent effects on SeV-induced IFN-β promoter activation. The expressions of NSP1, NSP3, NSP12, NSP13, NSP14, ORF3, ORF6, and M significantly inhibited SeV-mediated IFN-β activation, whereas NSP2 and S protein exhibited the opposite effects (Fig. 2c). Moreover, NSP1, NSP3, NSP12, NSP13, NSP14, ORF3, ORF6, E, and M were able

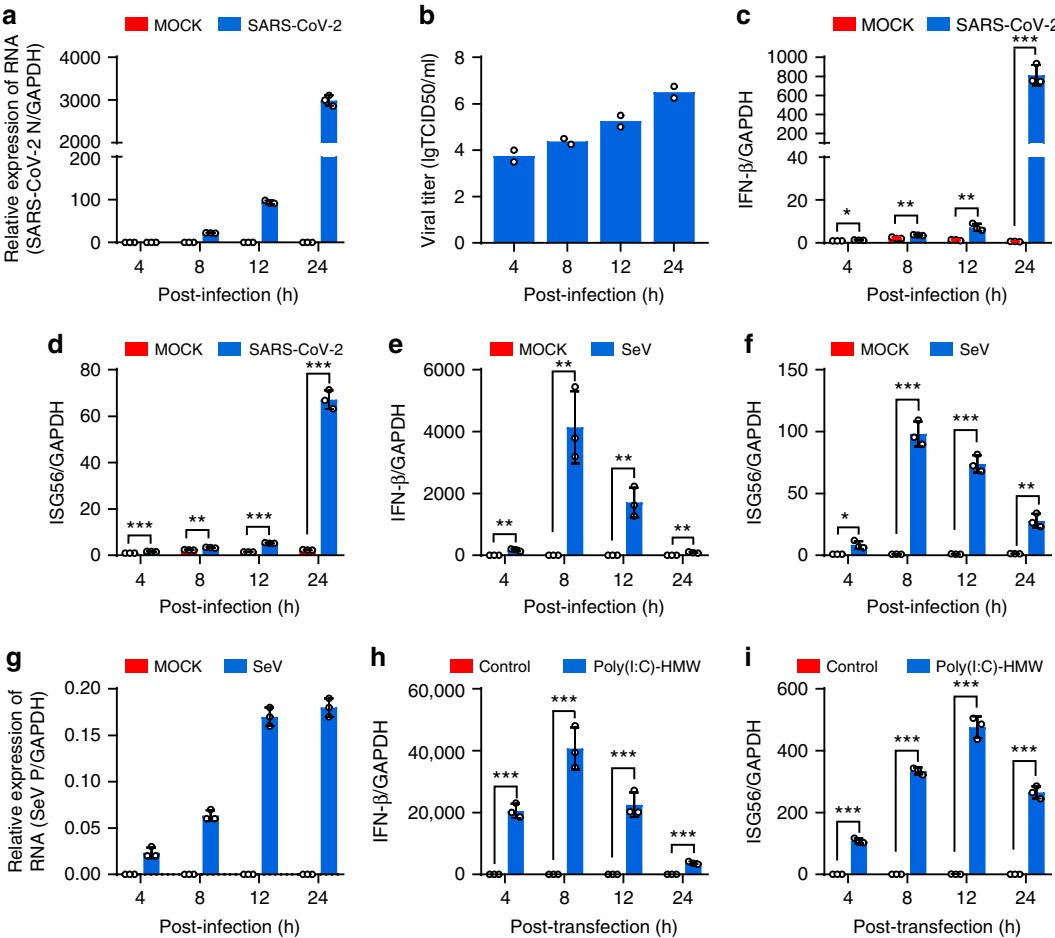

**Fig. 1 SARS-CoV-2 infection induces the expression of antiviral genes. a** Calu-3 cells were mock-infected or infected with SARS-CoV-2 at an MOI of 0.5. At 4, 8, 12, and 24 h after infection, total RNA extracted from cells was evaluated by quantitative real-time PCR (qRT-PCR) using SYBR green method. The data are expressed as fold changes of the RNA levels of the viral *N* gene relative to the *GAPDH* control. **b** Calu-3 cells were mock-infected or infected with SARS-CoV-2 at an MOI of 0.5. At 4, 8, 12, and 24 h after infection, supernatants were collected, and viral titers were detected by using $TCID_{50}$ assay. **c**, **d** Cells from **a** were collected, and total RNA extracted from the cells was evaluated by qRT-PCR using SYBR green method. The data are expressed as fold change of the *IFNB* mRNA **c** and *ISG56* mRNA **d** levels relative to the *GAPDH* control. **e–g** Calu-3 cells were mock-infected or infected with Sendai virus. Cells were harvested and analyzed as described in **c** and **d**. **h**, **i** Calu-3 cells were transfected with high molecular weight poly(I:C) (poly(I:C)-HMW). Cells were harvested at indicated times and analyzed by qRT-PCR. All experiments were done at least twice, and one representative is shown. Error bars indicate SD of technical triplicates. *$P < 0.05$, **$P < 0.01$, and ***$P < 0.001$, two-tailed Student's *t*-test. Source data are provided as a Source Data file.

to recapitulate their inhibitory activity when IFN-β promoter activity was stimulated upon the overexpression of RIG-IN (the constitutively active N-terminal domains of RIG-I) or MDA5 (Fig. 2d, e). The expression levels of SeV protein, RIG-IN, and MDA5 were shown in Fig. 2f–h. These results suggest that the SARS-CoV-2 proteins may play pleiotropic roles in regulating host innate immune response.

**SARS-CoV-2 ORF6 inhibits IFN-β activation**. Considering that SARS-CoV-2 ORF6 shares the least sequence similarity with SARS-CoV ORF6 (Supplementary Fig. 1a, b)[1], which was shown to counteract host antiviral response at multiple steps[29,31], we focused on the function of SARS-CoV-2 ORF6. Immuno-fluorescence experiments showed that ORF6 was predominantly localized in the cytoplasm and partially colocalized with the Golgi apparatus and endoplasmic reticulum markers (Supplementary Fig. 1c). In a dose-dependent manner, ORF6 inhibited IFN-β promoter activation induced by both SeV and the high molecular weight poly(I:C), which are thought to stimulate RIG-I and MDA5 signaling pathway, respectively (Fig. 3a, b). Further, we examined at which step of the signaling cascade the ORF6

blocks the antiviral innate immune response. We cotransfected increasing amounts of ORF6 expression plasmids with plasmids encoding key signaling proteins involved in innate antiviral response and determined the activation of the IFN-β promoter. As shown in Fig. 3c–f, overexpression of ORF6 inhibited RIG-IN, MDA5, MAVS, and IRF3-5D (a constitutively active IRF3 mutant)-triggered IFN promoter activation in a dose-dependent manner. These results demonstrated that ORF6 inhibited IFN-β production at the level of or downstream of IRF3 activation. Moreover, SARS-CoV-2 ORF6 and SARS-CoV ORF6 showed comparable inhibitory effects on MAVS and IRF3-induced IFN-β promoter activation (Fig. 3g, h). Finally, immunofluorescence analyses showed that SeV-induced IRF3 nuclear translocation was prevented in cells overexpressing ORF6 (Fig. 3i), corroborating that ORF6 blocks IRF3 activation.

**ORF6 suppresses IRF3 activation via its C-terminus**. Because the C-terminus of SARS-CoV ORF6 is critical for its antagonistic activity[29], we then examined the role of the C-terminus of SARS-CoV-2 in IFN inhibition. Three ORF6 variants harboring C terminal mutations, including ORF6-M1 (a.a. 49–52 substituted

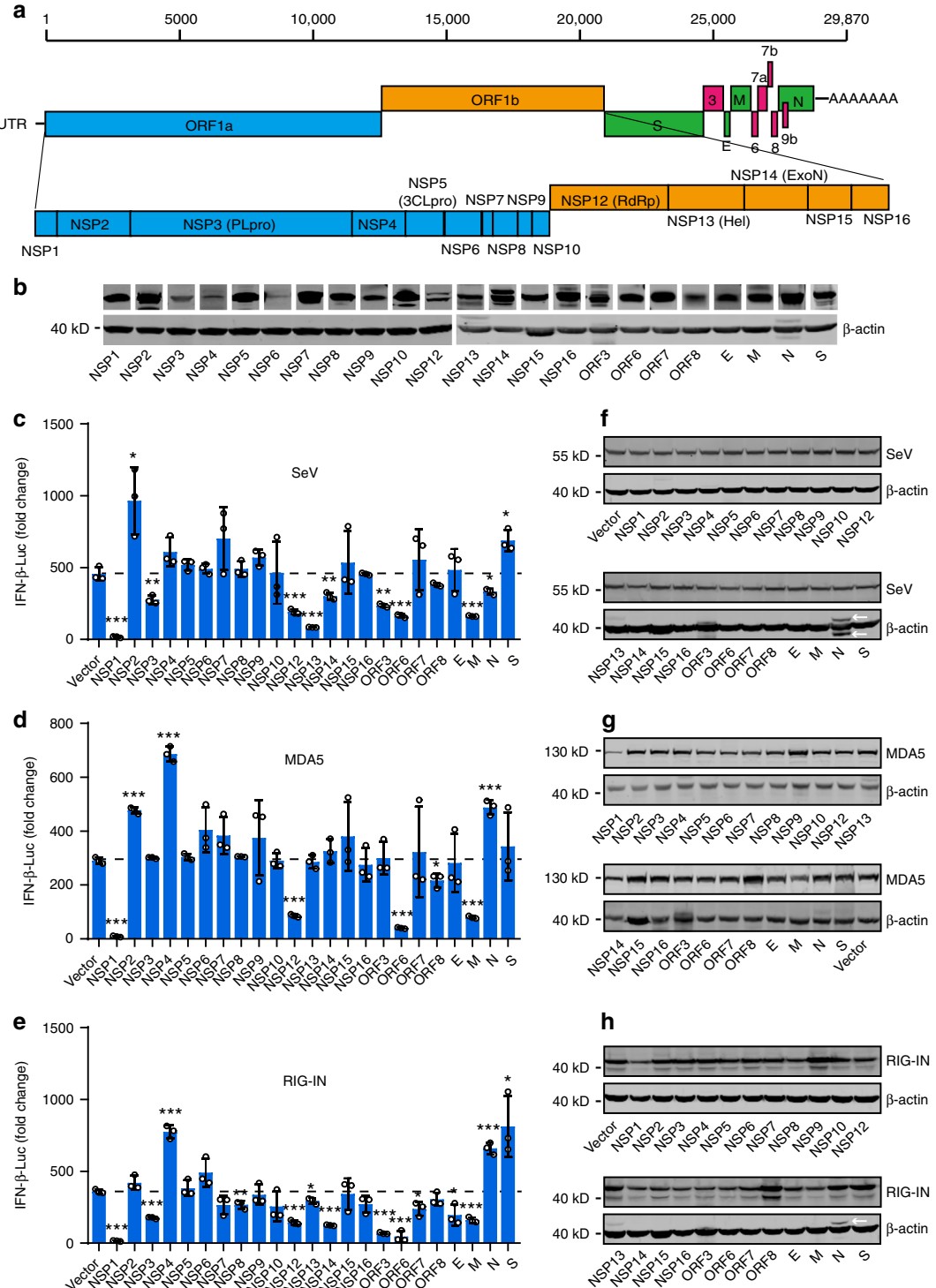

**Fig. 2 Identification of viral proteins perturbing IFN-β production. a** Schematic diagrams of the SARS-CoV-2 genome. The genome includes 5'UTR-ORF1a-ORF1b-S-ORF3-E-M-ORF6-ORF7 (7a and 7b)-ORF8-N-3'UTR in order. Fifteen nonstructural proteins, four structural proteins, and four accessory proteins were delineated. **b** Protein expressions of SARS-CoV-2 genes. HEK293T cells were transfected with 500 ng plasmid in 24-well plates. Protein expressions were detected by Western blot. β-actin was used as a loading control. **c** Effect of SARS-CoV-2 proteins on SeV-induced IFN-β promoter activation. HEK293T cells were transfected with an IFN-β reporter plasmid, along with a control plasmid or with plasmids expressing the indicated SARS-CoV-2 proteins. At 24 h post-transfection, cells were infected with SeV for 12 h, and luciferase activity was measured. **d, e** Effects of SARS-CoV-2 proteins on RIG-IN and MDA5-induced IFN-β promoter activation. HEK293T cells were transfected with IFN-β promoter plasmid, along with a control plasmid or with plasmids expressing the indicated SARS-CoV-2 proteins, together with a plasmid expressing RIG-IN **d** or MDA5 **e**. At 24 h post-transfection, cells were harvested and luciferase activity was measured. **f–h** Expressions levels of SARS-CoV-2 protein, RIG-IN, and MDA5. Lysates of cells from **c–e** were subjected to Western blot analysis. Arrows indicate remnants of blots for SARS-CoV-2 proteins. All experiments were done at least twice, and one representative is shown. Error bars indicate SD of technical triplicates. $*P < 0.05$, $**P < 0.01$, and $***P < 0.001$, two-tailed Student's $t$-test. Source data are provided as a Source Data file.

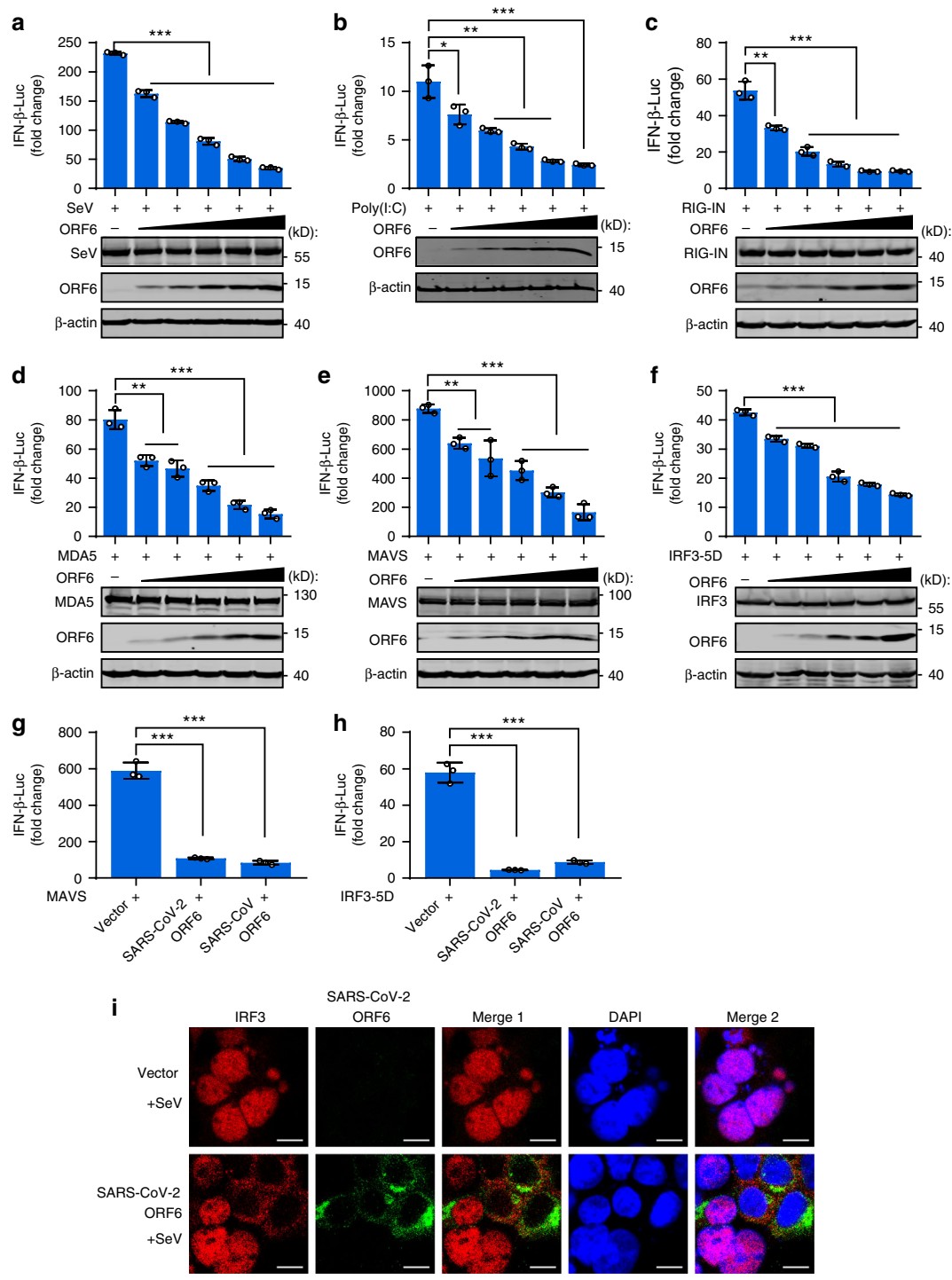

with alanines), ORF6-M2 (a.a.53–55 substituted with alanines), and ORF6-M3 (a.a. 56–61 substituted with alanines), were generated (Fig. 4a). Then, the effect of wildtype and ORF6 mutants on the IFN promoter activation was assayed. We found that overexpression of ORF6-M1 inhibited IFN-β promoter activation triggered by RIG-IN, MDA5, MAVS, and IRF3-5D to a comparable or lower level to that observed in the overexpression of wildtype ORF6 (Fig. 4b–e). In contrast, ORF6-M2 and ORF6-M3 exhibited severely impaired inhibitory activity. Moreover, SeV-induced IRF3 nuclear translocation was impeded by the overexpression of wildtype and the M1 form of ORF6, while ORF6-M2 and ORF6-M3 had no effects (Supplementary Fig. 2). Therefore, we conclude that the C-terminal tail of SARS-CoV-2

ORF6 from a.a. 53 to 61 are essential for its antagonistic activity, whereas a.a. 49 to 52 are dispensable.

**ORF6 inhibits the ISRE and ISG56 promoter's activation.** We further examined whether SARS-CoV-2 proteins affect downstream signaling of type I IFN. To do this, cells were cotransfected with the vector plasmid or with plasmids expressing SARS-CoV-2 proteins, along with an interferon-stimulated response element (ISRE) reporter plasmid. After 24 h, cells were treated with 800 U/ml IFN-β for 12 h and then subjected to dual luciferase assay. This assay revealed that NSP1, NSP3, NSP13, NSP14, ORF6, ORF8, N, and S proteins inhibited IFN-β-induced ISRE promoter activity, while NSP6, NSP9, NSP12, and E proteins showed stimulatory effects

**Fig. 3 SARS-CoV-2 ORF6 inhibits RIG-I-like signaling pathways. a** Effects of SARS-CoV-2 ORF6 on SeV-induced IFN-β promoter activation. HEK293T cells were transfected with an IFN-β reported plasmid, along with a control plasmid or with increasing amounts plasmids expressing ORF6. Cells were infected with SeV for 12 h and assayed for luciferase activity. **b** Effects of SARS-CoV-2 ORF6 on poly(I:C)-induced IFN-β promoter activation. HEK293T cells were transfected as described in **a**. At 24 h post-infected, cells were transfected with high molecular weight poly(I:C) (poly(I:C)-HMW) for 12 h and assayed for luciferase activity. **c–f** Effects of ORF6 on RIG-IN, MDA5, MAVS, or IRF3-induced IFN-β promoter activation. HEK293T cells were transfected with an IFN-β reporter plasmid, along with a control plasmid or with increasing amount plasmids expressing ORF6, together with plasmids expressing RIG-IN (**c**), MDA5 (**d**), MAVS (**e**), or IRF3-5D (**f**). At 24 h post-transfection, luciferase activity was measured. **g, h** Effect of SARS-CoV ORF6 and SARS-CoV-2 ORF6 on MAVS and IRF3-5D-induced IFN-β promoter activation. 293T cells were transfected with IFN-β reporter plasmid, along with a control plasmid or plasmids expressing SARS-CoV ORF6 or SARS-CoV-2 ORF6, together with plasmids expressing MAVS (**g**) or IRF3-5D (**h**). At 24 h post-transfection, luciferase activity was measured. **i** Confocal immunofluorescence imaging of IRF3 and SARS-CoV-2 ORF6. HEK293 cells were transfected with a control plasmid or a plasmid expressing SARS-CoV-2 ORF6. At 24 h of post-infection, cells were infected with SeV. At 4 h of post-infection, cells were stained with indicated antibodies and subjected to immunofluorescence analyses. Red: IRF3 antibody signal; Green: ORF6 signal; Blue: DAPI (nuclei staining). Merge 1 and Merge 2 indicate the merged red and green channels and the merged red, green, and blue channels, respectively. Scale bar, 10 μm. All experiments were done at least twice, and one representative is shown. Error bars indicate SD of technical triplicates. *$P < 0.05$, **$P < 0.01$, and ***$P < 0.001$, two-tailed Student's $t$-test. Source data are provided as a Source Data file.

(Fig. 5a). We next focus on ORF6. Overexpression of ORF6 inhibited luciferase expression from both ISRE and ISG56 promoter in a dose-dependent manner (Fig. 5b, c). Because expression from ISRE or ISG56 promoter after IFN-β treatment depends on IFN receptor signaling, these data demonstrated that ORF6 antagonizes signaling downstream of IFN. Further, SARS-CoV ORF6 and SARS-CoV-2 ORF6 showed a comparable inhibitory effect on the IFN-β-induced ISRE or ISG56 promoter activation (Fig. 5d, e).

**ORF6 inhibits STAT1 nuclear translocation via its C-terminus.** We then investigated the mechanism by which ORF6 inhibits IFN signaling. After binding to its receptor, IFN-β activates the Jak-STAT pathway, in which the Jak1 and Tyk2 kinases phosphorylate STAT1 and STAT2, triggering their dimerization and nuclear translocation[24]. In 293T cells, IFN-induced STAT1 phosphorylation remains intact in the presence of SARS-CoV and SARS-CoV-2 ORF6 (Fig. 6a). In contrast, expression of SOCS1, a well-established inhibitor of Jak kinases, substantially inhibited STAT1 phosphorylation (Fig. 6a). These data suggested that ORF6 does not interfere with signaling cascade upstream of STAT1 phosphorylation. Similar results were observed in Vero cells, which are deficient in type I IFN genes but retain the IFN receptor (Supplementary Fig. 3).

Having observed that ORF6 inhibited ISRE/ISG56 promoter activation while not affecting STAT1 phosphorylation, we then asked whether ORF6 overexpression regulates the translocation of STAT1 from the cytoplasm to the nucleus. Cells were transfected with plasmids expressing SARS-CoV or SARS-CoV-2 ORF6. After 24 h, cells were treated with IFN-β for 30 min, and the localization of STAT1 was analyzed. Immunofluorescence analyses showed that cells expressing either SARS-CoV ORF6 or SARS-CoV-2 ORF6 displayed rare STAT1 nuclear localization, whereas ORF6-null cells showed substantial STAT1 nuclear distribution (Fig. 6b, c), indicating that SARS-CoV and SARS-CoV-2 ORF6 inhibited IFN-β-triggered STAT1 nuclear translocation. Finally, we detected the effect of SARS-CoV-2 ORF6 C-terminal variants on STAT1 activation. We found that, to a comparable extent, both ORF6-M1 and wildtype ORF6 inhibited IFN-induced ISRE/ISG56 promoter activation and translocation (Fig. 7a–c), whereas failed to affect IFN-stimulated STAT1 phosphorylation (Supplementary Fig. 4). ORF6-M2 and ORF6-M3 exerted no effects in those assays. Moreover, immunofluorescence analyses with an antibody specific to phospho-STAT1 showed similar results that were observed in Fig. 7c (Supplementary Fig. 4b). Together, these data suggested that the C-terminal tail a.a. 53–61 are essential for ORF6's activity in antagonizing STAT1 nuclear translocation but not phosphorylation.

**SARS-CoV-2 is sensitive to IFN-β treatment.** Finally, we examined the effect of IFN-β treatment on SARS-CoV-2 infection. Calu-3 cells were pretreated with 100 or 500 U/ml of recombinant human IFN-β for 18 h to trigger IFN response, whose activation was then verified by the upregulated expression of ISG genes *ISG54* and *ISG56* (Fig. 8a, b). Then, cells were infected with SARS-CoV-2 at an MOI of 0.5 for 24 h. We found that IFN-β treatment decreased the amount of viral transcripts and the production of replicative viruses in a dose-dependent manner (Fig. 8c, d). Collectively, the results suggest that SARS-CoV-2 is sensitive to IFN-β treatment.

## Discussion
The viral antagonism of host innate immune response is critical for virus replication and often determines the outcomes of the infection. The evasion of host immune surveillance will give rise to the uncurbed viral replication, which could cause hyperactive host proinflammatory response, termed as hypercytokinemia or cytokine storm, and eventually lead to detrimental outcomes[32]. In the severe cases of COVID-19, hypercytokinemia and acute respiratory distress syndrome (ARDS) were observed[33–35], while the underlying mechanism remains unclear. In the current study, we revealed that SARS-CoV-2 induced an aberrant type-I IFN response in cultured cells, as the expressions of IFN-β and ISG56 were barely induced early during viral infection, while surged at late time points. This delayed antiviral response may provide a window for virus replication. Indeed, large amounts of viral transcripts were observed before the IFN induction in SARS-CoV-2-infected cells. Consonant with this, high viral loads were detected in COVID-19 cases soon after symptom onset[36]. Thus, we postulate that the lack of timely and adequate antiviral response may be central to the COVID-19 pathogenesis.

It is known that SARS-CoV has developed multiple strategies to antagonize the host antiviral response[37]. Among those, ORF6 was thought to play a critical role as this protein limits both IFN production and downstream signaling[29,31]. Of note, ORF6 of SARS-CoV-2 shares the least sequence similarity with that of SARS-CoV (~66%), and the disparity is mainly observed in the C-terminal sequence. Moreover, SARS-CoV-2 has a two amino acid truncation in its ORF6 terminus tail as compared to SARS-CoV. Considering that the C-terminus tail of SARS-CoV ORF6 is required for its antagonistic activity[29], we inferred ORF6 maintains equivalent function in SARS-CoV-2. In the current study, we revealed that ORF6 of SARS-CoV-2 and SARS-CoV exhibited similar cellular distribution and comparable ability in inhibiting IRF3 activation and STAT1 nuclear translocation. Thus, although genetically changed, SARS-CoV-2 ORF6 retained its full ability in antagonizing host innate immune response, suggesting that this

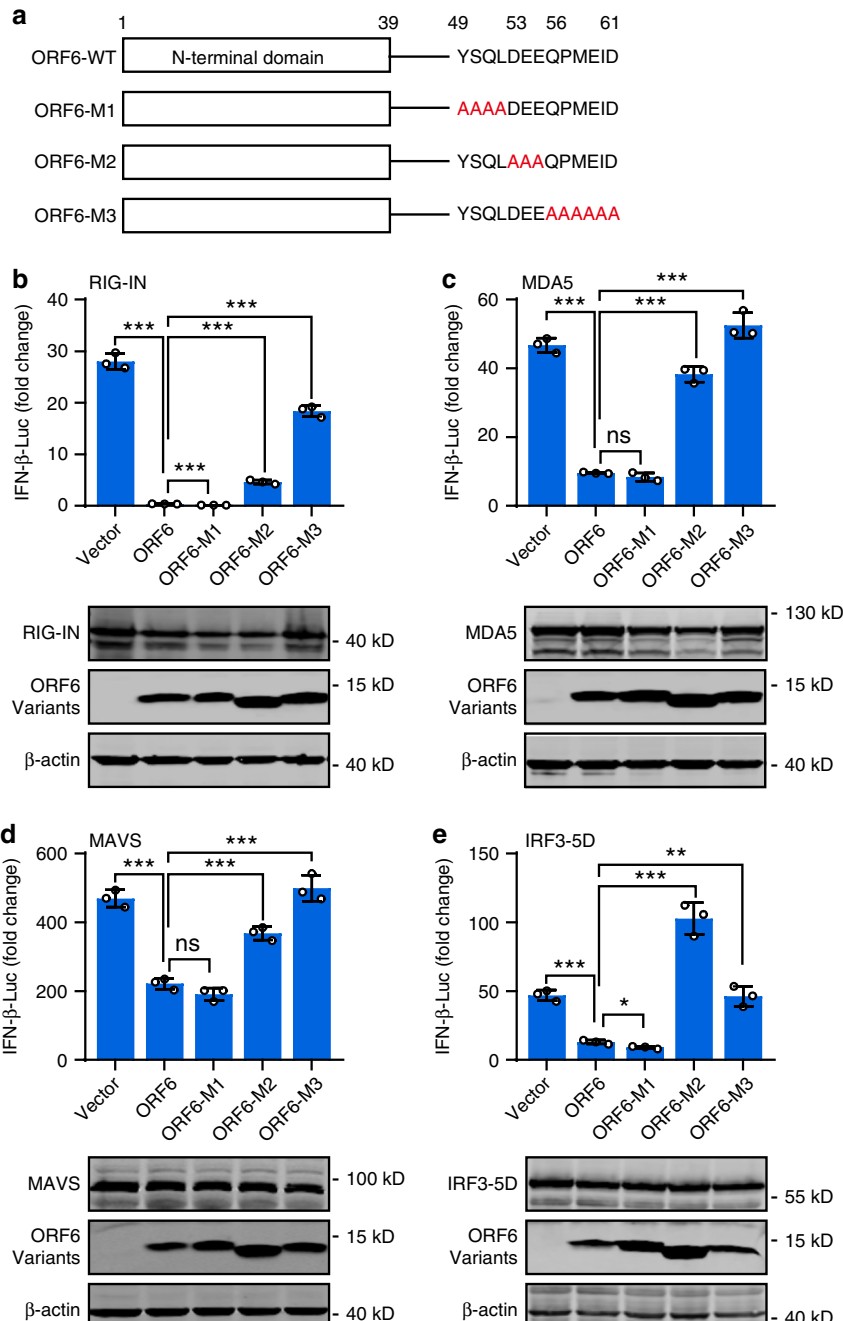

**Fig. 4 ORF6 antagonizes innate immune response via its C terminus. a** Schematic diagram of SARS-CoV-2 ORF6 variants. ORF6-WT: wildtype, ORF6-M1: amino acids 49–52 were substituted with alanines, ORF6-M2: amino acids 53–55 were substituted with alanines; ORF6-M3: amino acids from 56 to 61 were substituted with alanines. **b–e** The effect of ORF6 mutants on IFN-β promoter activation. HEK293T cells were transfected with an IFN-β reporter plasmid, along with a control plasmid or plasmids expressing wildtype ORF6 or indicated ORF6 variants, together with plasmids expressing RIG-IN (**b**), MDA5 (**c**), MAVS (**d**), or IRF3-5D (**e**). At 24 h after transfection, luciferase activity was measured. Protein expression levels were detected by Western blot. All experiments were done at least twice, and one representative is shown. Error bars indicate SD of technical triplicates. *$P < 0.05$, **$P < 0.01$, and ***$P < 0.001$, ns not significant, two-tailed Student's $t$-test. Source data are provided as a Source Data file.

antagonistic function is evolutionarily important. Of note, the last several amino acids of the ORF6 C-terminus tail, DEEPMELDYP in SARS-CoV[29], and DEEQPMEID in SARS-CoV-2, are indispensable for ORF6's function in blocking IRF3 and STAT1 activation. Therefore, we hypothesized that these amino acids, which are enriched with negatively charged residues, may provide an interface that mediates ORF6's interaction with host proteins, and thus exert its antagonistic activity. This sequence could be a therapeutic target candidate because a small blocking peptide against this motif could potentially mitigate SARS-CoV-2's virulence. Recently, in an effort to map SARS-CoV-2-human protein–protein interaction (PPIs), Gordon et al. identified that ORF6 interacts with NUP98 and RAE1, which form a nuclear pore complex[38]. It is possible that ORF6 blocks IRF3 and STAT1 nuclear translocation by interacting with these nuclear pore proteins. This hypothesis awaits further investigations.

Other than ORF6, several SARS-CoV proteins, including NSP1, ORF3b, M, N, and others, may act as IFN antagonists[37].

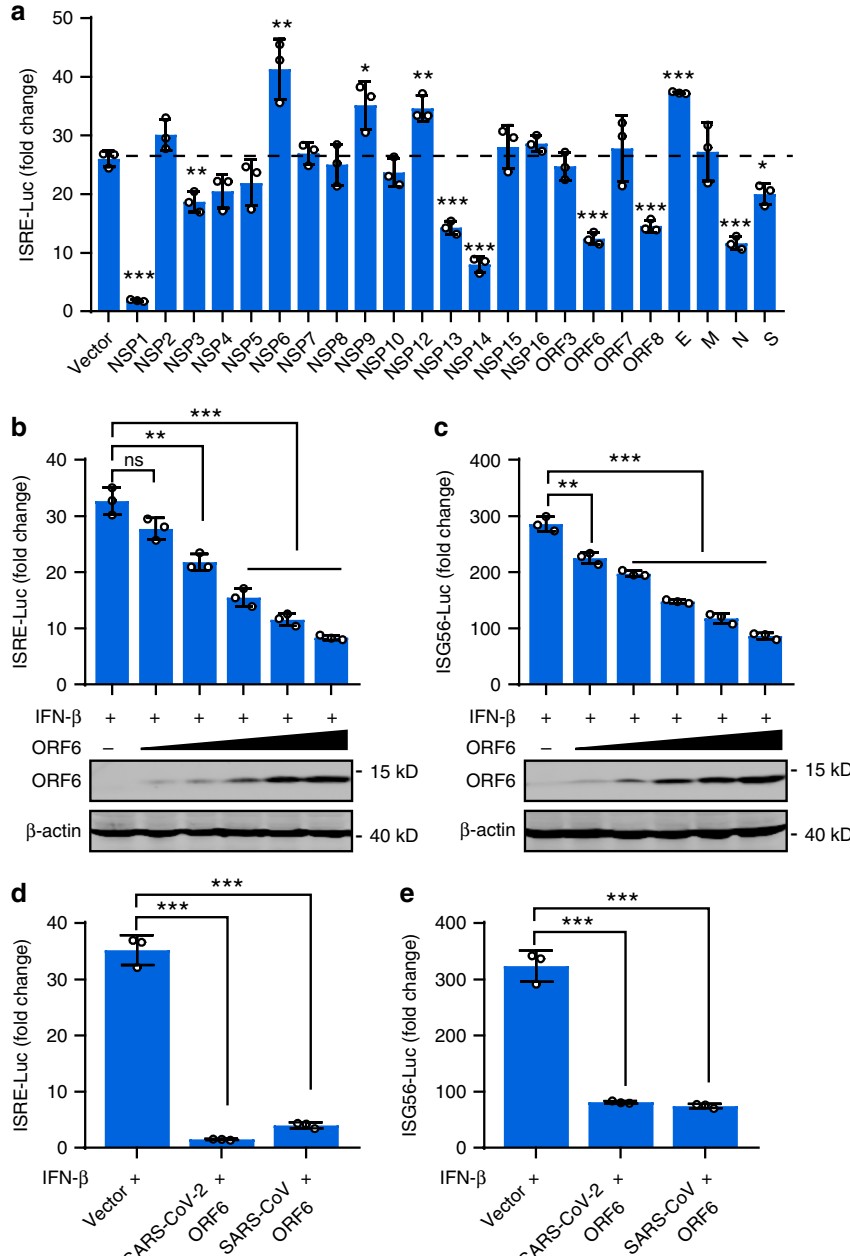

**Fig. 5 ORF6 inhibits ISRE promoter activation. a** The effects of SARS-CoV-2 proteins on ISRE-promoter activation. HEK293T cells were transfected with an ISRE reporter plasmid, along with a control plasmid or with plasmids expressing the indicated SARS-CoV-2 proteins. At 24 h post-transfection, cells were treated with 800 U/ml of human IFN-β for 12 h, and luciferase activity was measured. **b**, **c** The effects of SARS-CoV-2 ORF6 on ISRE and ISG56 promoter activity. HEK293T cells were transfected with am ISRE reporter plasmid (**b**) and an ISG56 reporter plasmid (**c**), along with a control plasmid or the increasing amounts of plasmids expressing SARS-CoV-2 ORF6. At 24 h post-transfection, cells were treated with 800 U/ml of human IFN-β for 12 h, and luciferase activity was measured. **d**, **e** Effect of SARS-CoV ORF6 and SARS-CoV-2 ORF6 on ISRE and ISG56 promoters. HEK293T cells were transfected with an ISRE reporter plasmid (**d**) or an ISG56 reporter plasmid (**e**), along with a control plasmid or plasmids expressing SARS-CoV ORF6 or SARS-CoV-2 ORF6. At 24 h post-transfection, cells were treated with 800 U/ml of human IFN-β for 12 h, and luciferase activity was measured. All experiments were done at least twice, and one representative is shown. Error bars indicate SD of technical triplicates. *$P < 0.05$, **$P < 0.01$, and ***$P < 0.001$, two-tailed Student's *t*-test. Source data are provided as a Source Data file.

Here, we found that NSP1 and M of SARS-CoV-2 also inhibited SeV-induced IFN production. Moreover, this inhibition effect was observed when overexpressing NSP12 and NSP13, which are enzymes responsible for genomic replication of coronaviruses[39]. Of note, in the PPI study[38], NSP13 was identified to interact with TBK1 and its adapter TBKBP1. Because TBK1 plays essential roles in innate antiviral response, we speculate that NSP13 could mitigate IFN production inhibiting TBK1 activity. Intriguingly,

NSP2 and S of SARS-CoV-2 showed a significant stimulatory effect on the IFN induction. This unexpected observation confounded the overall effect of the SARS-CoV proteins on the innate antiviral immune response. Recent studies suggested that SARS-CoV-2 could induce expression of multiple ISGs[35,40], which is barely detected in SARS-CoV infection[41]; we speculate that the immuno-stimulatory effects of SARS-CoV-2 proteins may contribute to this induction.

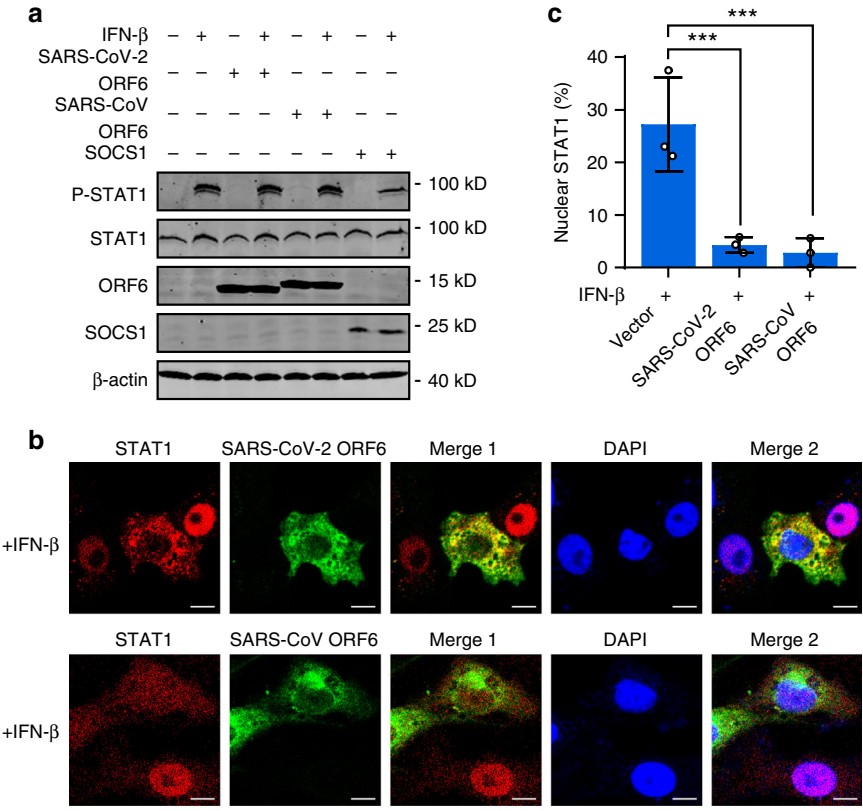

**Fig. 6 ORF6 inhibits STAT1 nuclear translocation but not phosphorylation. a** Effect of SARS-CoV ORF6 and SARS-CoV-2 ORF6 on IFN-β-induced phosphorylation of STAT1. HEK293T cells were transfected with a control plasmid or with plasmids expressing SARS-CoV ORF6, SARS-CoV-2 ORF6, or SOCS1. At 24 h after transfection, cells were left untreated or treated with 1000 U/ml IFN-β for 30 min. The phosphorylation of STAT1 was detected by Western blot analyses. **b** Effect of SARS-CoV ORF6 and SARS-CoV-2 ORF6 on IFN-β-induced nuclear translocation of STAT1. Vero cells were transfected with plasmids expressing SARS-CoV ORF6 and SARS-CoV-2 ORF6. At 24 h after transfection, cells were treated with 1000 U/ml IFN-β for 30 min and stained with indicated antibodies. Merge 1 and Merge 2 indicate the merged red and green channels and the merged red, green, and blue channels, respectively. Scale bar, 10 μm. **c** Quantitation of the nuclear translocation of STAT1. All experiments were done at least twice, and one representative is shown. Error bars indicate SD of technical triplicates. ***$P < 0.001$, two-tailed Student's $t$-test. Source data are provided as a Source Data file.

In agreement with recent findings[42,43], we found that SARS-CoV-2 is sensitive to IFN pretreatment, suggesting that IFN therapy could be an option for COVID-19 treatment. It is of interest to identify ISGs that directly and specifically inhibit SARS-CoV-2 infection and replication. Overall, our study characterized the interplay between SARS-CoV-2 and host innate immunity, and provided mechanistic insight in the immune evasion of SARS-CoV-2 mediated by viral proteins. These findings could advance our understandings of the pathogenesis of SARS-CoV-2.

## Methods

**Cell lines and viruses**. Human 293T (ATCC, #CCL-11268;) cells, 293 (ATCC, #CRL-1573), Calu-3 (ATCC, #HTB-55;), HeLa (ATCC, #CCL-2;), and Vero (ATCC, #CCL-81,) cells was cultured in Dulbecco's modified Eagle's medium (Invitrogen, Carlsbad, CA) supplemented with 10% heat-inactivated fetal bovine serum (FBS) (HyClone, Logan, UT), 100 U/ml penicillin, and 100 U/ml streptomycin at 37 °C in a 5% $CO_2$ humidified atmosphere. Low passage HeLa and Vero cells after directly purchasing from ATCC were used, and all cells were tested for mycoplasma-free. The SARS-CoV-2 virus was isolated from respiratory samples of confirmed COVID-19 patients by inoculating onto Vero cells[1] and was propagated in Vero cells and used in this study. Cells were infected with SARS-CoV-2 at a multiplicity of infection (MOI) of 0.5. Unbound virus was washed away after 1 h, and cells were then cultured with fresh medium supplemented with 2% FBS. All experiments with the SARS-CoV-2 virus were conducted in the BSL-3 laboratory.

**Plasmids and antibodies**. The 23 genes of SARS-CoV-2 (IPBCAMS-WH-01/2019 strain, no. EPI_ISL_402123) were optimized by Gene Designer 1.0 and cloned to vector pCMV6-entry expression vector with the FLAG-tag or HA-tag at C-terminus. Plasmids Flag-RIG-I, Flag-RIG-IN, Flag-MDA5, HA-MAVS,

pGL3-IFN-β–Luc, IRF3-5D-Flag, and pRL-TK have been described elsewhere[42]. The mutated variants of the SARS-CoV-2 ORF6-tagged Flag were constructed by using a Site-Directed Mutagenesis Kit (Stratagene, La Jolla, CA). All variants were confirmed by subsequent sequencing.

The antibodies used in this research were: Flag antibody from Sigma-Aldrich (1:4000, Cat# F3165); β-actin antibody from Sigma–Aldrich (1:4000, Cat# A5441); HA antibody from Sigma–Aldrich (1:10,000, Cat# H9658); STAT1 antibody from Cell Signaling technology (1:1000, Cat# 9172); P-STAT1 antibody from Thermo Fisher (1:1000, Cat# 700349); Sev antibody from MBL (1:2000, PD029C1). Dual-Luciferase® Reporter Assay System was purchased from Progema (Madison, WI). IRDye 800-labeled IgG and IRDye 680-labeled IgG secondary antibodies were purchased from Li-Cor Biosciences (Lincoln, NE).

**Fifty percent tissue culture infectious dose (TCID$_{50}$) assays**. Samples were stored at −80 °C and repeatedly freeze-thaw three times before being processed for determination. Vero cells in 96-well plates were cultured overnight with 80% confluency. Using dilution blocks, samples were serially diluted 10-fold from $10^{-1}$ to $10^{-8}$ in opti-MEM. 100 μl/well of each dilution were placed onto the Vero cells in octuplicate and incubated at 37 °C with 5% $CO_2$ for 1 h. Then the culture supernate was replaced with 1% BSA of opti-MEM and incubated for 4 days. Then the cytopathic effect (CPE) was evaluated under a microscope and recorded.

**Reporter assays**. 293T cells cultured in 24-well plates were transfected with a control plasmid or plasmids expressing of RIG-IN, MDA5, MAVS, or IRF3-5D, along with luciferase reporter plasmids or plasmids expressing viral proteins. Cells were harvested, and cell lysates were used to determine luciferase using a Dual Luciferase Reporter Assay System (Promega). The firefly luciferase activities were normalized to Renilla luciferase activities[44].

**Immunofluorescence**. Cells were washed with PBS buffer and fixed with 4% formalin. Then cells were permeabilized with 0.5% Triton X-100. After cells were washed with PBS, they were blocked and stained with primary antibodies, followed by staining

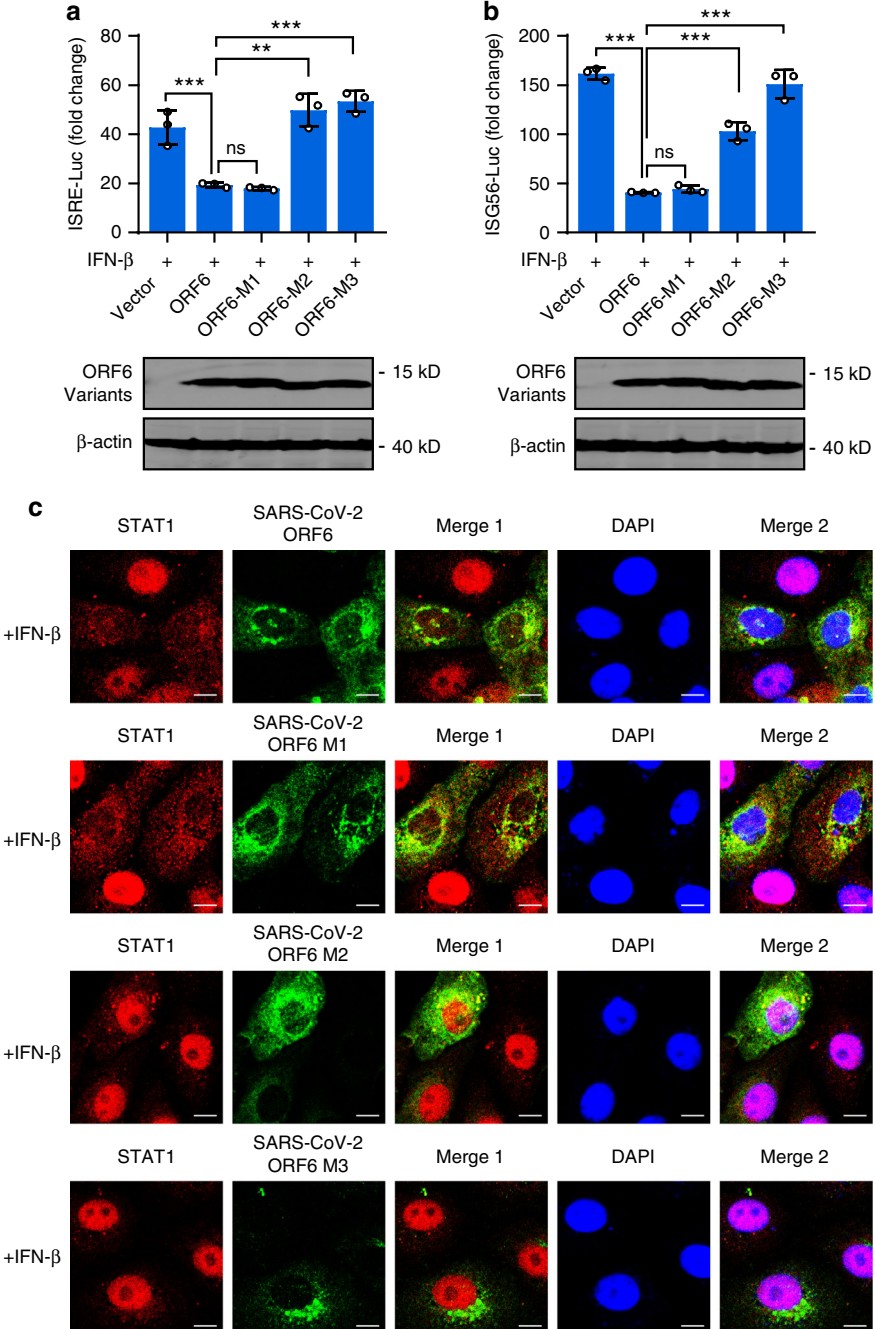

**Fig. 7 ORF6 inhibits STAT1 nuclear translocation via its C-terminus. a, b** Effect of SARS-CoV-2 ORF6 and its mutants on IFN-β-induced ISRE- and ISG56-promoter activation. HEK293T cells were transfected with an ISRE reporter plasmid (**a**) or an ISG56 reporter plasmid (**b**), along with a control plasmid or plasmids expressing wildtype or indicated SARS-CoV-2 ORF6 variants. At 24 h post-transfection, cells were treated with 800 U/ml IFN-β for 12 h, and luciferase activity was measured. Protein expression levels were detected by Western blot analyses. **c** Effect of SARS-CoV-2 ORF6 and its variants on IFN-β-induced STAT1 nuclear translocation. Vero cells were transfected with plasmids expressing wildtype ORF6 or indicated ORF6 variants. At 24 h post-transfection, cells were treated with 1000 U/ml of human IFN-β for 30 min and stained with indicated antibodies. Merge 1 and Merge 2 indicate the merged red and green channels and the merged red, green, and blue channels, respectively. Scale bar, 10 μm. All experiments were done at least twice, and one representative is shown. Error bars indicate SD of technical triplicates, **$P < 0.01$, and ***$P < 0.001$, ns not significant, two-tailed Student's $t$-test. Source data are provided as a Source Data file.

with an Alexa Fluor 488 secondary antibody[45]. Nuclei were stained with DAPI (Sigma). The antibodies used in this research were: IRF3 antibody from Cell Signaling technology (1:200, Cat# 11904); STAT1 antibody from Cell Signaling technology (1:400, Cat# 14994); P-STAT1 antibody from Cell Signaling technology (1:400, Cat# 9167); Calnexin antibody from Cell Signaling technology (1:50, Cat# 2679); GolgiB1 antibody from Sigma-Aldrich (1:500, Cat# HPA011008). Fluorescence images were obtained and analyzed using a laser scanning confocal microscope (Leica TCS SP5).

**Quantitative real-time PCR analysis**. Total RNA was extracted by using TRIzol reagent (Invitrogen, Carlsbad, CA) and reverse transcript to cDNA by M-MLV Reverse Transcriptase (Promega, Madison, WI). cDNAs were prepared for the real-time PCR by using TB Green Premix Ex (Takara, Kusatsu, Shiga). The Primer sequences of *IFNB, ISG56, ISG54, P* gene of SeV were provided in Supplementary Table 1. The Primer sequence specific for the SARS-CoV-2 was available from J.W. upon request.

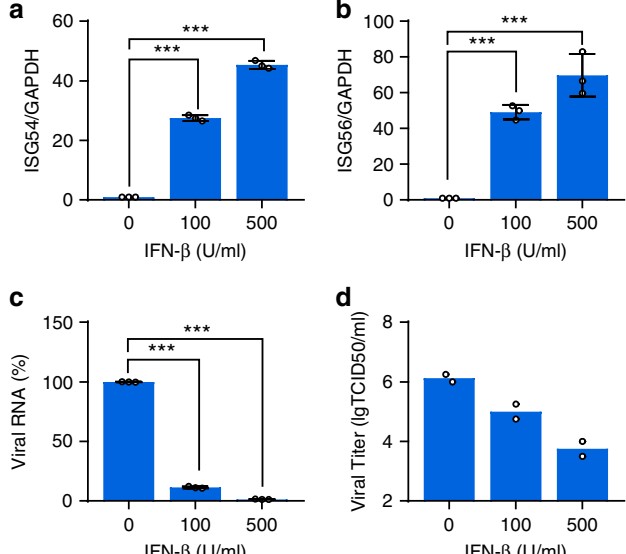

**Fig. 8 IFN-β inhibits SARS-CoV-2 replication. a, b** Expressions of *ISG54* and *ISG56* upon IFN-β treatment. Calu-3 cells were left untreated or treated with 100 or 500 U/ml of recombinant human IFN-β for 18 h. Total RNA was extracted, and the *ISG54* and *ISG56* mRNA were detected by qRT-PCR. **c** Susceptibility of SARS-CoV-2 to IFN-β treatment. Calu-3 cells were left untreated or pretreated with 100 or 500 U/ml human IFN-β for 18 h, and then cells were mock-infected or infected with SARS-CoV-2 at an MOI of 0.5. After 24 h post-infection, SARS-CoV-2 RNA was detected by qRT-PCR using SYBR green. **d** Viral titer assessment. Supernatants from **c** were harvested and subjected to $TCID_{50}$ analyses for measuring viral titers. All experiments were done at least twice, and one representative is shown. Error bars indicate SD of technical triplicates. ***$P < 0.001$(***), two-tailed Student's *t*-test. Source data are provided as a Source Data file.

**Statistics**. The two-tailed Student's *t*-test was used for two-group comparisons. The values *$P < 0.05$, **$P < 0.01$, and ***$P < 0.001$ were considered significant. ns stands for not significant.

**Reporting summary**. Further information on research design is available in the Nature Research Reporting Summary linked to this article.

## Data availability

All other data are included in the Supplemental Information or available from the authors upon reasonable requests. Source data are provided with this paper.

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

## Acknowledgements

This work was supported by grants from the National Major Sciences & Technology Project for Control and Prevention of Major Infectious Diseases in China (2018ZX10301401 to X.L. and Z. Zhou, 2018ZX10733403 to Z.X.), the National Natural Science Foundation of China (81930063, 81971948, 31670169 to J.W., X.L., and Z. Zhou), National Key R&D Program of China (2020YFA0707600 to X.L.), Chinese Academy of Medical Sciences (CAMS) Innovation Fund for Medical Sciences (2016-I2M-1-014, 2016-I2M-1-005 to J.W. and X.L.), and the Beijing Advanced Innovation Center for Genomics (ICG).

## Author contributions

Project conception: X.L., Z.X., and J.W.; Experimental design: X.L., L.R., Z. Zhou, Z.X., J.W.; Experimental work: X.D., R.M., W.W., X.X., C.W., Z.T., Y.W., L.L.; Data analysis: X.L., L.R., Z. Zhou, Z.X., Z. Zhao and J.W., Writing original draft: X.L., Z. Zhou, Z.X., and J.W.; Writing review and editing, J.W., X.L., Z. Zhou., Z.X., F.G.; All authors reviewed the manuscript.

## Competing interests

The authors declare no competing interests.
