## [Peer Review File · Nature Communications]

REVIEWER COMMENTS

Reviewer #1 (Viral immunity, interferon response) (Remarks to the Author):

This is an interesting paper that examines the interaction of the new pandemic SARS-CoV-2 virus with the interferon system (IFN). The authors show that SARS-CoV-2 induces IFN following viral RNA synthesis and they show that viral RNA synthesis is impaired by IFN treatment. A systematic investigation of the properties of the individual proteins/ORFs of SARS-CoV-2 to block IFN production and/or IFN signalling is presented. The authors conclude that the SARS-CoV-2 ORF6 is a major antagonist of both IFN production and IFN signalling. The former is also shown to depend upon a sequence at the C-terminus of SARS-CoV-2.

Overall, there is a lot of useful information in this paper, but several of the conclusions are incompletely justified.

Specific comments:

1. A major weakness of the paper is that RNA levels are used as a surrogate for virus production. For example, Figure 1A is reported as showing that SARS-CoV-2 replicates well in Calu3 cells. All these data show are that viral RNA levels have increased, not that the virus has productively replicated. Similarly, in Figure 7C the effect of IFN treatment on virus is merely showing that IFN down-regulates viral RNA production. Virus production needs to be quantified by direct technique, such as plaque assays.
2. The argument that expression of IFN- β is "severely delayed" in SARS-CoV-2 infection is flawed. The key question is "how quickly would you expect IFN- β to be induced"? In most viruses the induction of IFN- β depends upon the production of PAMPs by viral RNA synthesis, and that is precisely what is seen in this paper. The idea that IFN- β induction is slow in response to SARS-CoV-2 infection is based on the single comparison to Sendai virus. Although no details are given about the prep of Sendai virus the authors used, most people use the "Cantell" strain, a prep that is very high in defective interfering (DI) particles; the DI particles act as a rapid inducer of IFN- β . In a normal Sendai infection (i.e. using a prep which lacks the DI particles), and indeed many other relatively slowly replicating viruses, IFN- β induction starts to occur at 15 – 18 hours.
3. In the experiments which examine the abilities of the SARS-CoV-2 proteins to block IFN- β induction only Sendai virus is used as a source of PAMP. This means that only the RIG-I pathway is examined; since coronaviruses are thought to activate mda5, the failure to try poly(I).poly(C) as a PAMP that activates mda5 is a missed opportunity.
4. Only ORF6 has been examined in a concentration-dependence assay; all of the data shown in Figure 2 apparently use a single plasmid concentration; the authors need to specify what that is, and need to comment on this.
5. The paper needs to be proofread, because as it stands it contains errors. The data shown for Figures 3G – I are a comparison between SARS-CoV-2 and the "original" SARS-CoV; the description of these experiments on p7 do not mention SARS-CoV. Similarly, on p8, the comparison between SARS-CoV-2 and the "original" SARS-CoV is described as "SARS-CoV-2 ORF6" and "SARS-CoV-2 ORF6".
6. The western blot data shown in Figures 5E and S2 are carried out using transient transfection and therefore need some sort of statement on transfection efficiency. In 293 cells most cells will be transfected and hence it is probably reasonable to conclude that STAT1 phosphorylation is not blocked

by SARS-CoV-2 ORF6 expression. However, Vero cells (Figure S2) are not that well transfected. A general consensus is 30% or less – this means that most of the detected STAT1 will be coming from untransfected cells, so any effect of SARS-CoV-2 ORF6 would be marginal.

7. The immunofluorescence data presented in Figures 5F and 5G are confusing. It is not possible to see STAT1 in cells which express SARS-CoV-2 ORF6 so it cannot be concluded that SARS-CoV-2 ORF6 blocks nuclear translocation. Can more convincing data be provided and can we see a "merge" that doesn't have the DAPI stain in it (this obscures any nuclear STAT1)?

Reviewer #2 (Influenza/viral responses, innate immunity) (Remarks to the Author):

The paper describes a screen of SARS2 viral ORFs for their ability to limit activation of the human IFN promoter. While a number of ORFs are found to have this inhibitory function, ORF6 is further characterised and mapped to inhibit at the level of IRF3 or below. Furthermore, ORF6 is taken forward and confirmed to behave like SARS1-ORF6 in inhibiting IFN signalling at the level of STAT1 translocation, and a region of ORF6 is mapped as being required for this activity.

The experiments seem to be generally well performed and described, and this is the first such screen for SARS2, clearly an important new virus. The speed of experimentation to get to this point is commendable, and identification of a few IFN antagonists encoded by SARS2 will be an important addition to the field and of interest to many. However, the work is still preliminary, and there are several areas where a few extra simple assays would have greatly improved the depth of understanding about this new virus and its IFN antagonist proteins. Such extra work would turn this manuscript into a much more valuable piece of work and thereby also have greater impact in influencing the field than the current manuscript.

Major points:

1. Why was only ORF6 taken forward for the ISRE assay? The manuscript could be more informative if the whole library of viral ORFs were also screened against the ISRE to have a full dataset on viral ORFs that inhibit IFN β promoter and ISRE. It would also have been great to see the other viral IFN-antagonist ORFs identified in the IFN β screen assayed to see which level of the cascade they inhibit (RIG-I, MDA5, MAVS, TBK1 etc) as was done for ORF6. This would add a lot to the manuscript.
2. ORF6 appears to inhibit STAT1 translocation (but not phosphorylation), dependent upon the M2 sequence (Fig. 5). Given that ORF6 inhibits IRF3 function (Fig 3), the authors should test whether IRF3 translocation to the nucleus (but not phosphorylation) is also impacted by ORF6 in an M2-dependent fashion.
3. Does SARS2 infection block IFN signalling? The ORF6 over-expression data should be complemented by infection data – do cells that are SARS2-infected exhibit similar block to STAT1 translocation in response to exogenous IFN?
4. The authors should validate with specific amino-acid substitutions their interesting hypothesis that it really is the PMExD motif impacting SARS2-ORF6 IFN-antagonistic function. This would provide important sequence mapping for surveillance efforts and provide a rational basis for interfering peptides (line 250-253).
5. Which cellular factors interact with ORF6 in the recent Nature paper from Nevan Krogan? This should be discussed, particularly if any of them are related to nuclear trafficking functions. Ideally, the authors should validate the interaction by co-IP and test if the M2 sequence (or PMExD motif) is

required for the interaction. This would provide important mechanistic insights for the reader.

6. Fig. 5E would be improved by a positive control for a viral protein that blocks STAT1 phosphorylation to show that transfection efficiency is high enough to see effects on endogenous protein phosphorylation (if transfection efficiency low, then unlikely to see effects).

Other points to address:

1. Some English language editing is required throughout.
2. Lines 156-159 – this sentence is unclear – are the authors trying to say the inhibition is similar to ORF6 or SARS-CoV-1? Please clarify.
3. Lines 193-196 are repetitive with lines 188-onwards. Please clarify.
4. IRF7 is mentioned in the methods, but I could not see it mentioned elsewhere – please clarify.
5. Co-IPs are mentioned in the methods, but no-where else. Clarify?
6. Figure legends should give an indication of independent replicate numbers, statistics and error bar meanings. Individual replicates should be plotted. Statistics are mentioned in the methods but not in the figures/legends.
7. For the ORFs identified to antagonise IFN induction in the Figure 2 screen, it would be informative to the reader if these ORFs were described in the context of their interactomes identified by the Krogan lab in the recent Nature paper.
8. In Fig 1D, what are the other bands in the actin blots for N and NSP16/ORF3 (as a few examples?). These should be discussed – are they remnants of blots for other factors (eg the viral ORFs)? Then please highlight. Figure 1B is weak as there is no loading control to compare across the different ORFs – please improve.
9. All western blots should have kDa markers indicated.
10. Figure 4B and C – please clarify the ORF6 western blot – these look identical, but slightly different exposure – are they really from different transfections (one RIG-I, one MDA-5)?

Response to Reviewers

We thank all reviewers for their interest in our work and their constructive comments to strengthen our study. Below we provide a point-by-point response (normal script) to each reviewer comment (italics).

Reviewer #1 (Viral immunity, interferon response) (Remarks to the Author):

This is an interesting paper that examines the interaction of the new pandemic SARS-CoV-2 virus with the interferon system (IFN). The authors show that SARS-CoV-2 induces IFN following viral RNA synthesis and they show that viral RNA synthesis is impaired by IFN treatment. A systematic investigation of the properties of the individual proteins/ORFs of SARS-CoV-2 to block IFN production and/or IFN signalling is presented. The authors conclude that the SARS-CoV-2 ORF6 is a major antagonist of both IFN production and IFN signalling. The former is also shown to depend upon a sequence at the C-terminus of SARS-CoV-2.

Overall, there is a lot of useful information in this paper, but several of the conclusions are incompletely justified.

- We thank the Reviewer for noting the importance of our work.

Specific comments:

1. A major weakness of the paper is that RNA levels are used as a surrogate for virus production. For example, Figure 1A is reported as showing that SARS-CoV-2 replicates well in Calu3 cells. All these data show are that viral RNA levels have increased, not that the virus has productively replicated. Similarly, in Figure 7C the effect of IFN treatment on virus is merely showing that IFN down-regulates viral RNA production. Virus production needs to be quantified by direct technique, such as plaque assays.

- As suggested by the Reviewer, we have determined virus titer in the supernatants by TCID₅₀ assays in the experiments related to Figure 1A and Figure 7C of the original manuscript. Please refer to the revised figure 1B and figure 8D.

2. The argument that expression of IFN- β is “severely delayed” in SARS-CoV-2 infection is flawed. The key question is “how quickly would you expect IFN- β to be induced”? In most viruses the induction of IFN- β depends upon the production of PAMPs by viral RNA synthesis, and that is precisely what is seen in this paper. The idea that IFN- β induction is slow in response to SARS-CoV-2 infection is based on the single comparison to Sendai virus. Although no details are given about the prep of Sendai virus the authors used, most people use the “Cantell” strain, a prep that is very high in defective interfering (DI) particles; the DI particles act as a rapid inducer of IFN- β . In a normal Sendai infection (i.e. using a prep which lacks the DI particles), and indeed many other relatively slowly replicating viruses, IFN- β induction starts to occur at 15 – 18 hours.

- We agree with the Reviewer that the statement “severely delayed” should be more rigorously tested. First, we would like to explain that SARS-CoV-2 is not a slowly replicating virus. In our experiment, number of SARS-CoV-2 *N* gene

transcripts exceeded host GAPDH transcript at 8 hpi. A recent paper (Bojkova et al, *Nature* 2020) also reported that “SARS-CoV-2 rapidly replicates in cells”. Next, to address the question “how quickly would you expect IFN- β to be induced”, we performed novel experiments to examine the kinetics of poly(I:C)-induced IFN- β expression. This experiment circumvented virus replication and showed that poly(I:C) transfection stimulated the expression of IFN- β and as early as 4 hpi and peaked at 8 hpi (revised Figure 1H and I). This kinetics is very similar to that observed in the SeV-induced IFN- β expression (revised Figure 1E and F). In contrast, upon SARS-CoV-2 infection, the expressions of IFN β were only marginally elevated until 12 hours postinfection (hpi) but were dramatically induced at 24 hpi (revised Figure 1C). Therefore, we came to the conclusion that SARS-CoV-2 infection stimulated delayed IFN production.

3. In the experiments which examine the abilities of the SARS-CoV-2 proteins to block IFN- β induction only Sendai virus is used as a source of PAMP. This means that only the RIG-I pathway is examined; since coronaviruses are thought to activate mda5, the failure to try poly(I).poly(C) as a PAMP that activates mda5 is a missed opportunity.

- As suggested by the Reviewer, we included a novel experiment, which used high molecular weight poly(I:C) to stimulate the MDA5 pathway. Please refer to the revised Figure 3B. Besides, we examined the effect of SARS-CoV-2 proteins on IFN β promoter activation upon overexpression of MDA5 (Revised Figure 2D).

4. Only ORF6 has been examined in a concentration-dependence assay; all of the data shown in Figure 2 apparently use a single plasmid concentration; the authors need to specify what that is, and need to comment on this.

- The plasmid concentration is 500 ng/well of 24-well plate. We have included this information in the revised figure legend. Please refer to line 407.

5. The paper needs to be proofread, because as it stands it contains errors. The data shown for Figures 3G – I are a comparison between SARS-CoV-2 and the “original” SARS-CoV; the description of these experiments on p7 do not mention SARS-CoV. Similarly, on p8, the comparison between SARS-CoV-2 and the “original” SARS-CoV is described as “SARS-CoV-2 ORF6” and “SARS-CoV-2 ORF6”.

- We apologize for the errors. Now we have carefully proofread the paper and corrected all mistakes. Please refer to lines 173-174 and 207.

6. The western blot data shown in Figures 5E and S2 are carried out using transient transfection and therefore need some sort of statement on transfection efficiency. In 293 cells most cells will be transfected and hence it is probably reasonable to conclude that STAT1 phosphorylation is not blocked by SARS-CoV-2 ORF6 expression. However, Vero cells (Figure S2) are not that well transfected. A general consensus is 30% or less – this means that most of the detected STAT1 will be coming from untransfected cells, so any effect of SARS-CoV-2 ORF6 would be marginal.

- To address the Reviewer's concern, we have repeated the experiment. Meanwhile, we included a positive control side by side: we transfected a plasmid expressing SOCS1, a well-established inhibitor of JAK kinases. We found that, although the expression level of SOCS1 is lower than SARS-CoV-2 ORF6 and all ORF6 mutants, SOCS1 substantially inhibited STAT1 phosphorylation. Again, ORF6 has no effects on STAT1 phosphorylation. Therefore, the failure in blocking STAT1 phosphorylation is not due to the low expression level of ORF6. Please refer to the revised Figures 6A and S4.

7. The immunofluorescence data presented in Figures 5F and 5G are confusing. It is not possible to see STAT1 in cells which express SARS-CoV-2 ORF6 so it cannot be concluded that SARS-CoV-2 ORF6 blocks nuclear translocation. Can more convincing data be provided and can we see a "merge" that doesn't have the DAPI stain in it (this obscures any nuclear STAT1)?

- As suggested by the Reviewer, we have updated the images showing apparent STAT1 signals. Moreover, we provided additional merged images without showing DAPI staining. Please refer to the revised Figures 6B and 7C.

Reviewer #2 (Influenza/viral responses, innate immunity) (Remarks to the Author):

The paper describes a screen of SARS2 viral ORFs for their ability to limit activation of the human IFN promoter. While a number of ORFs are found to have this inhibitory function, ORF6 is further characterised and mapped to inhibit at the level of IRF3 or below. Furthermore, ORF6 is taken forward and confirmed to behave like SARS1-ORF6 in inhibiting IFN signalling at the level of STAT1 translocation, and a region of ORF6 is mapped as being required for this activity.

The experiments seem to be generally well performed and described, and this is the first such screen for SARS2, clearly an important new virus. The speed of experimentation to get to this point is commendable, and identification of a few IFN antagonists encoded by SARS2 will be an important addition to the field and of interest to many. However, the work is still preliminary, and there are several areas where a few extra simple assays would have greatly improved the depth of understanding about this new virus and its IFN antagonist proteins. Such extra work would turn this manuscript into a much more valuable piece of work and thereby also have greater impact in influencing the field than the current manuscript.

- We thank the Reviewer for the encouraging comments.

Major points:

1. Why was only ORF6 taken forward for the ISRE assay? The manuscript could be more informative if the whole library of viral ORFs were also screened against the ISRE to have a full dataset on viral ORFs that inhibit IFN β promoter and ISRE. It would also have been great to see the other viral IFN-antagonist ORFs identified in

the IFN β screen assayed to see which level of the cascade they inhibit (RIG-I, MDA5, MAVS, TBK1 etc) as was done for ORF6. This would add a lot to the manuscript.

- As suggested by the Reviewer, we have screened the effect of all viral proteins on IFN- β -stimulated ISRE activity. Please refer to the revised Figure 5A. Moreover, we performed a systematic assay on viral proteins' effect on IFN- β promoter activity induced by RIG-I and MDA5, two critical pattern recognition receptors in innate immune signaling. Please refer to the revised Figures 2D, G, E, and H.

2. ORF6 appears to inhibit STAT1 translocation (but not phosphorylation), dependent upon the M2 sequence (Fig. 5). Given that ORF6 inhibits IRF3 function (Fig 3), the authors should test whether IRF3 translocation to the nucleus (but not phosphorylation) is also impacted by ORF6 in an M2-dependent fashion.

- To address this issue, we generated an additional C-terminus mutation of ORF6 and tested whether IRF3 translocation to the nucleus is impacted by wildtype and ORF6 mutations. We found that SeV-induced IRF3 nuclear translocation was impeded by the overexpression of wildtype and ORF6-M1 (a.a. 49-52 substituted with alanines). In contrast, ORF6-M2 (a.a. 53-55 substituted with alanines) and ORF6-M3 (a.a. 56-61 substituted by alanines) had no effects, suggesting that C-terminal a.a. 53-61 of ORF6 are important for its antagonistic activity. Please refer to the revised figures 3I and S2 and lines 175-178 and 190-192.

3. Does SARS2 infection block IFN signalling? The ORF6 over-expression data should be complemented by infection data – do cells that are SARS2-infected exhibit similar block to STAT1 translocation in response to exogenous IFN?

- Because exogenous IFN β treatment severely impairs SARS-CoV-2 replication, it is hard to examine if SARS-CoV-2 infection affect STAT1 translocation stimulated by exogenous IFN β treatment. To address this, we observed the STAT1 nuclear translocation during SARS-CoV-2 infection without IFN β treatment. We found that, even though SARS-CoV-2 actively replicated, STAT1 fully retained in the cytoplasm, suggesting that SARS-CoV-2 infection does stimulate STAT nuclear translocation. Please refer to the data below.

4. The authors should validate with specific amino-acid substitutions their interesting hypothesis that it really is the PMExD motif impacting SARS2-ORF6 IFN-antagonistic function. This would provide important sequence mapping for

surveillance efforts and provide a rational basis for interfering peptides (line 250-253).

- In the original manuscript, we have made two ORF6 mutants, i.e., ORF6-M1 (a.a. 49-52 substituted with alanines) and ORF6-M2 (a.a. 56-61 substituted by alanines). Of note, the C-terminal a.a. 53-55 were not included in the substitution assays. To more comprehensively mapping the C-terminal a.a. of ORF6, we made an extra mutant whose a.a. 53-55 were substituted by alanines. We found that a.a. 53-55 are also important for ORF6's antagonistic activity. Therefore, the a.a. 53-61 (DEEQPMEID), rather than the previously proposed PMExD motif, are indispensable for ORF6's function in blocking IRF3 and STAT1 activation. This region is enriched with negatively charged residues, which may cooperatively act to provide an interface for protein-protein interaction. We have revised the discussion accordingly. Please refer to line 277-280 in the revised manuscript.

5. Which cellular factors interact with ORF6 in the recent Nature paper from Nevan Krogan? This should be discussed, particularly if any of them are related to nuclear trafficking functions. Ideally, the authors should validate the interaction by co-IP and test if the M2 sequence (or PMExD motif) is required for the interaction. This would provide important mechanistic insights for the reader.

- This is a valuable suggestion, but we feel that it is out of scope of the current manuscript. We have discussed the potential interaction between ORF6 and host factors identified in Nevan Krogan's paper. Please refer to lines 285-287.

6. Fig. 5E would be improved by a positive control for a viral protein that blocks STAT1 phosphorylation to show that transfection efficiency is high enough to see effects on endogenous protein phosphorylation (if transfection efficiency low, then unlikely to see effects).

- We have included a positive control side by side: we transfected a plasmid expressing SOCS1, a well-established inhibitor of JAK kinases. We found that, although the expression level of SOCS1 is lower than SARS-CoV-2 ORF6 and all ORF6 mutants, SOCS1 substantially inhibited STAT1 phosphorylation. Again, ORF6 has no effects on STAT1 phosphorylation. Therefore, the failure in blocking STAT1 phosphorylation is not due to the low expression level of ORF6. Please refer to the revised Figure 6A and S4.

Other points to address:

1. Some English language editing is required throughout.

- We have carefully proofread the manuscript and improved our English language with the help of a native English speaker.

2. Lines 156-159 – this sentence is unclear – are the authors trying to say the inhibition is similar to ORF6 or SARS-CoV-1? Please clarify.

- The Reviewer is correct. We have now improved this sentence. Please refer to line 173-175 in the revised manuscript.

3. Lines 193-196 are repetitive with lines 188-onwards. Please clarify.

- We have removed these repetitive words. Please refer to line 211-213 in the revised manuscript.

4. IRF7 is mentioned in the methods, but I could not see it mentioned elsewhere – please clarify.

- We have removed IRF7 from the Methods. Please refer to line 326 in the revised manuscript.

5. Co-IPs are mentioned in the methods, but no-where else. Clarify?

- We have removed the description of Co-IP from the Methods.

6. Figure legends should give an indication of independent replicate numbers, statistics and error bar meanings. Individual replicates should be plotted. Statistics are mentioned in the methods but not in the figures/legends.

- As suggested, we have improved the figure legends by providing information on statistical analysis. Individual replicates have been plotted where it is appropriate. Please refer to the revised figure and figure legends.

7. For the ORFs identified to antagonise IFN induction in the Figure 2 screen, it would be informative to the reader if these ORFs were described in the context of their interactomes identified by the Krogan lab in the recent Nature paper.

- As suggested, we have discussed those ORFs in the context of the interactomes identified in the recent Nature paper. Please refer to lines 285-287 in the revised manuscript.

8. In Fig 1D, what are the other bands in the actin blots for N and NSP16/ORF3 (as a few examples?). These should be discussed – are they remnants of blots for other factors (eg the viral ORFs)? Then please highlight. Figure 1B is weak as there is no loading control to compare across the different ORFs – please improve.

- Yes, in revised Fig 2F and H, the bands are remnants of blots for other SARS-CoV-2 proteins. We have highlighted these bands and commented in the figure legend. Please refer to the revised Figure 2 F and H and the related Figure legends.
- We have provided loading control in the revised Figure 2B.

9. All western blots should have kDa markers indicated.

- As suggested, we have labeled molecular weight in all western blots data. Please refer to the revised figures.

10. Figure 4B and C – please clarify the ORF6 western blot – these look identical, but slightly different exposure – are they really from different transfections (one RIG-I, one MDA-5)?

- Figure 4B and C are different western blots. Please refer to the raw data below.
- These two panels have been replaced in the revised manuscript as we have new data on ORF6 mutants.

REVIEWERS' COMMENTS:

Reviewer #1 (Remarks to the Author):

I have read the response to the reviewer's comments and I am satisfied that the authors have made reasonable efforts to address these comments. As far as I am concerned they have answered all my queries and I think the manuscript is much improved.

Reviewer #2 (Remarks to the Author):

All of my original comments have been satisfactorily addressed

REVIEWERS' COMMENTS:

Reviewer #1 (Remarks to the Author):

I have read the response to the reviewer's comments and I am satisfied that the authors have made reasonable efforts to address these comments. As far as I am concerned they have answered all my queries and I think the manuscript is much improved

Response: We thank the reviewer for reviewing our work again. We are delighted that all concerns have been successfully addressed.

Reviewer #2 (Remarks to the Author):

All of my original comments have been satisfactorily addressed

Response: We thank the reviewer for reviewing our work again. We are delighted that all concerns have been successfully addressed.